# Operational Definitions of Polypharmacy and Their Association with All-Cause Hospitalization Risk: A Conceptual Framework Using Administrative Databases

**DOI:** 10.3390/pharmacy13010015

**Published:** 2025-02-02

**Authors:** Stefano Scotti, Lorenza Scotti, Federica Galimberti, Sining Xie, Manuela Casula, Elena Olmastroni

**Affiliations:** 1Epidemiology and Preventive Pharmacology Service (SEFAP), Department of Pharmacological and Biomolecular Sciences, University of Milan, 20122 Milan, Italy; stefano.scotti@unimi.it (S.S.); sining.xie@unimi.it (S.X.); elena.olmastroni@unimi.it (E.O.); 2IRCCS MultiMedica, 20099 Sesto San Giovanni, Italy; federica.galimberti@multimedica.it; 3Department of Translational Medicine, University of Piemonte Orientale UPO, 13100 Novara, Italy; lorenza.scotti@uniupo.it

**Keywords:** polypharmacy, drug exposure, operational definitions, administrative databases, epidemiologic methods, hospitalization risk

## Abstract

Polypharmacy, defined as the concurrent use of multiple medications, increases the risk of various adverse outcomes. However, the variability in definitions across the literature contributes to substantial heterogeneity. Building on the published literature, this study aimed to identify a set of operational definitions of polypharmacy applicable to administrative databases and to assess their association with all-cause hospitalization. Data from the pharmacy refill and hospitalization databases of the Local Health Unit (LHU) of Bergamo, Lombardy, were analyzed. Patients aged ≥40 with at least one reimbursed drug prescription in 2017 were included. Prescription coverage was evaluated using total defined daily doses (DDDs), and all-cause hospitalizations from January to June 2018 were considered. Definitions explored included (i) the WHO’s criterion of ≥5 medications by ATC fourth-level code; (ii) the exclusion of prescriptions usually for short-term treatments; and (iii) drugs with cumulative annual DDD ≥ 60. Approaches were assessed annually, quarterly, and monthly, and logistic regression was used to estimate odds ratios (ORs) for hospitalization risk. Among 431,620 patients, the DDD ≥ 60 definition showed the least variability (20.47–21.16%) and identified an older more complex cohort. All definitions showed a dose-dependent association with hospitalization risk. Different definitions of polypharmacy result in varying prevalence, with DDD ≥ 60 being the most consistent. A patient-centric approach is crucial to assess the appropriateness of polypharmacy.

## 1. Introduction

Polypharmacy, the concurrent use of multiple medications, is an increasingly prevalent phenomenon, particularly among older adults and individuals with chronic health conditions [1]. This arises from the need to manage multiple comorbidities, the involvement of multiple healthcare providers, the lack of coordinated care, and the rising availability of over-the-counter medications. This is exacerbated by an increasing medical specialization within healthcare systems, leading to multiple treatment regimens, often without any reconciliation process. Even when several concomitant drug therapies are necessary in a subject affected by several diseases, the pharmacological burden may result in organ impairment, drug–drug interactions (DDIs), drug–pathology interactions, other adverse reactions (ADRs), medication non-adherence, morbidity, mortality, poor quality of life, and increased healthcare costs due to unnecessary drug expenses [2,3]. This alarming public health concern is frequent, and the prevalence is increasing over time. A nationwide analysis in Poland revealed a polypharmacy prevalence of 14.9% among citizens aged 50–64 years in 2019 [4]. Similarly, a trend analysis conducted in the United States demonstrated an increase in polypharmacy prevalence from 10.5% in 1999/2000 to 17.0% in 2017–2018 among subjects aged 40–64 years [5].

The elderly population is certainly among the most susceptible subpopulations, as the burden of chronic diseases and drugs prescribed increases with age, and patients older than 65 years with multimorbidity are most vulnerable to the adverse effects of polypharmacy [6]. A recent analysis of a large European cohort has found polypharmacy (defined as the concurrent use of five or more medications) to be present in 32.1% of citizens aged 65 years or above, ranging from 26.3 to 39.9% across 17 countries [7]. In Italy, 68.0% of patients aged ≥65 years used five or more drugs during the year 2023 [8], while 28.5% used at least 10 different drugs [8].

The term polypharmacy is widely used in the literature, often referring to five or more drugs taken daily [9], while hyper-polypharmacy, or excessive polypharmacy, commonly refers to the concurrent use of 10 or more daily medications [9]. However, these general definitions have been translated into multiple operational definitions in several studies, without consensus on how to quantitatively measure it [9,10,11]. Moreover, it is unclear how much polypharmacy variability due to different definitions may affect prevalence estimates and the association with clinical outcomes. In this context, building on the published literature, we aimed to identify a set of operational definitions of polypharmacy applicable to administrative databases and to assess their association with the risk of all-cause hospitalization using real-world data

## 2. Materials and Methods

### 2.1. Data Sources and Study Population

The clinical setting of the study was general practice. Data were obtained from the EDU.RE.DRUG study, which have been described in detail elsewhere [12]. For this study, administrative data from the local health unit (LHU) of Bergamo in the Lombardy region were used. In particular, the archive of the LHU’s residents assisted from the Italian National Health Service (NHS) contains demographic variables (sex, date of birth, date of death); the pharmacy refill database contains data about all prescriptions of drugs reimbursed by the NHS (i.e., delivery date, Anatomical Therapeutic Chemical [ATC] code, national marketing authorization code, number of drug packages prescribed for each prescription); and the hospital discharge archive records contains information on the admission date and primary and secondary diagnoses of all hospitalizations at public or private hospitals.

Compliance with national and European laws on personal data was guaranteed by the LHU through the generation of unique anonymous codes for each patient. The study population included all community-dwelling adult patients (≥40 years) with at least one reimbursed drug prescription during the year 2017 and alive at the end of 2018.

### 2.2. Data Management

In the Italian administrative databases, prescriptions are characterized by the national marketing authorization code, from which the dosage and number of dosage units can be derived. This information, together with the defined daily dose (DDD), allows us to estimate the number of days covered by each prescription. For selected subjects, we extracted the entire pharmaceutical history relating to the period 1 January 2017–31 December 2017, including prescriptions with the dispensing date in 2016 and with at least one day covered in 2017.

Before calculating polypharmacy, we performed additional processing on the pharmaceutical dataset to more closely simulate what happens in everyday life. First, all drugs within the same therapeutic class (ATC 4th level) were considered interchangeable; second, when a drug was dispensed before the coverage (calculated through DDDs) of the previous prescription ended, utilization of the new medication was assumed to begin the day after the end of the previous prescription, shifting the entire coverage period of the subsequent prescription (Appendix A). Then, a supply diary was created for each patient day by stringing together consecutive fills of each medication class based on dispensing dates and days’ supply. Finally, for the calculation of polypharmacy, we counted the drugs in each month/quarter in which at least one day was covered by the prescription (Appendix A).

For the enrolled cohort, we investigated any hospitalization due to all causes that occurred in the period of 1 January 2018–1 June 2018, distinguishing between hospitalized and non-hospitalized patients (outcome). To prevent the potential misinterpretation of non-hospitalized patients, we excluded from the analysis subjects without a hospitalization in the first half of 2018 but reporting a hospitalization in the second half. Regardless of the outcome, for each subject in the cohort, we also assessed the number of any prior hospitalizations during the period of 2014–2017 (based on data availability).

### 2.3. Polypharmacy Definitions

To test different approaches in the assessment of polypharmacy, we considered a recent systematic review of the literature [9] and, among those that could be applied to administrative healthcare databases, we selected three operational definitions of polypharmacy differing in the drugs included (Table 1) as follows:We applied the WHO’s proposed definition of polypharmacy, i.e., the presence of 5 or more different medications (according to the ATC code). This definition will be referred to as “All ATC” [13];We excluded prescriptions of drugs usually associated with short-term treatments [4] (stomatological, anti-constipation, antibacterials for systemic use, antimycotics for systemic use, antivirals for systemic use as serums and immunoglobulins, vaccines, ectoparasiticides, dermatological, various). This definition will be referred to as “Chronic ATC”;We considered only drugs (ATC 4th level) with a cumulative annual DDDs ≥ 60 [14]. This definition will be referred to as “DDD ≥ 60”.

### 2.4. Statistical Analysis

Demographic statistics of the sample were summarized using frequencies and percentages or means and standard deviations as appropriate.

For each definition, polypharmacy was defined as the presence of ≥5 drugs for at least one day covered by each drug (ATC 4th level) in the pre-specified time window. Subjects were also classified based on the number of drugs prescribed (1–4 no polypharmacy, 5, 6, 7, 8, 9, or ≥10).

All the approaches were evaluated within pre-specified time windows of one year, three months, and one month. To report monthly and quarterly measures of polypharmacy over the year for each subject, we first calculated the number of different drugs administered every month or quarter within one year, and then we classified the subject as “on polypharmacy” if the maximum number of dispensed drugs among the 12 months or 4 quarters was at least 5 (Appendix A).

Unadjusted logistic regression models were performed to estimate odds ratios (ORs) and 95% confidence intervals [95% CIs] for the association between polypharmacy or the number of dispensed drugs and the risk of hospitalization for all causes. Then, multivariable logistic regression models were adjusted by sex, age (on 31 December 2017), and the number of previous hospitalizations.

Several sensitivity analyses have been performed. First, to validate the proposed approach in which we shifted the overlapping prescriptions, the polypharmacy was calculated also without considering the overlap of drugs within the same therapeutic class nor considering the coverage of the prescription. Second, we also evaluate the variability of the results obtained when the monthly and quarterly measures of polypharmacy over the year were calculated as the average (instead of the maximum) number of dispensed drugs or when the monthly measures of polypharmacy over the year are calculated as having at least 50%, 80%, or 100% of the follow-up period covered by 5 or more medications (which correspond to having 6, 10, or 12 months with at least 5 drugs dispensed each, respectively). Third, we evaluated the association between polypharmacy and the risk of hospitalization for all causes, considering only hospitalizations longer than 7 days.

## 3. Results

### 3.1. Prevalence of Polypharmacy

The cohort included 431,620 residents in the LHU of Bergamo, aged at least 40 years and with at least one drug dispensed in the year 2017, corresponding to 67.7% of the total LHU population aged ≥40 years in 2017 (N = 637,350 according to national demographic data).

The top five most frequently prescribed drugs/drug classes in this cohort were proton pump inhibitors (149,574 subjects), vitamin D and analogs (110,845 subjects), 3-hydroxy-3-methylglutaryl-CoA (HMG-CoA) reductase inhibitors (94,384 subjects), combinations of penicillins, including beta-lactamase inhibitors (92,486 subjects), and fluoroquinolones (80,205 subjects) (see Appendix A).

Figure 1 shows the prevalence of polypharmacy (five or more drugs) according to the three definitions applied in the three time windows. The selection of only drugs with at least 60 DDDs/year led to estimates with limited variability depending on the time window assessed (range 20.47–21.16%, 0.69 percentage points), while the definition using “all ATC” determined the greatest variability (range 8.74 percentage points).

In the sensitivity analysis in which the prevalence of patients in polypharmacy was calculated without considering the overlap of drugs within the same therapeutic class nor considering the coverage of the prescription (Appendix A), the percentages did not change when the evaluation was performed on the year (due to the fact that in the main analysis, we also considered prescriptions dispensed in the year 2016 that ended in 2017) and on the quarter (probably due to the fact that most of the prescriptions have almost a bimonthly duration, average duration 47.7 [SD 41.5] days). In contrast, a clear reduction in prevalence was observed when polypharmacy was assessed on a monthly basis (specifically, minus 16 percentage points for the All ATC definition; minus 15 percentage points for the Chronic ATC definition; minus 11 percentage points for the DDD ≥ 60 definition).

Results using the average number of dispensed drugs instead of the maximum are reported in Appendix A. This approach reduced the prevalence calculated on quarterly and on monthly bases, with a greater impact for the definition using all ATC (−8 and −11 percentage points, respectively) and lower impact for the definition using only drugs with at least 60 DDD (−1 and −3 percentage points, respectively). When we calculated the prevalence of patients in polypharmacy only taking into account patients having at least 50%, 80%, or 100% of their follow-up period covered by five or more medications, we observed that (i) all the prevalences were extremely lower, since the definition of polypharmacy in this scenario is much more conservative; (ii) all the prevalences tend to decrease with the increase in time period required to be covered by at least five drugs, as expected; (iii) within the same time period, all definitions applied led to similar prevalences (Appendix A). Table 2 reported the distribution of patients by polypharmacy classes based on the polypharmacy definition and the time window.

The definition “DDD ≥ 60” seems to identify a cohort of subjects who tend to be older; more complex (as suggested by the higher prevalence of hospitalizations; Table 3); and also characterized by a higher average number of dispensed drugs in 2017 compared to the cohorts identified with the other two definitions of polypharmacy.

### 3.2. Risk of All-Cause Hospitalization

The logistic regression analysis showed that even after adjustment, all the definitions were associated with a statistically significant increased risk of all-cause hospitalization (Table 4). The definition “DDD ≥ 60” appeared to be associated with a lower variability (adjusted odd ratios) in the estimates across models conducted over different time windows. Moreover, this definition was also associated with the lowest risk estimates compared with the others.

In Figure 2, the adjusted estimates for the association between polypharmacy classes and the risk of hospitalization for all causes are reported. A dose-dependent increase in risk can be observed as the number of the dispensed drugs increases for all definitions of polypharmacy considered, regardless of the time windows. Moreover, within the same time window, the ORs tend to decrease for each polypharmacy class when the definition of polypharmacy involves chronic treatments.

Finally, we evaluated in a sensitivity analysis the association between polypharmacy classes and the risk of all-cause hospitalization, considering only hospitalizations longer than 7 days (Appendix A). All results obtained were confirmed with higher estimates, as expected, given that only the most severe hospitalizations are considered as outcomes.

## 4. Discussion

### 4.1. Key Findings

Our analysis applied various operational definitions of polypharmacy in the general population of a Northern Italian area using administrative databases. Different operational definitions yielded different prevalence estimates; however, a clear dose-dependent increase in all-cause hospitalization risk was observed with all the definitions. The lack of a standard operational definition leads to multiple methodologies and approaches for polypharmacy calculation, influenced by population, medical areas, geographic setting, and data availability. Globally, these factors account for a high variability in reported prevalence estimates of polypharmacy, ranging between 10% and 90% [15]. Our approach was based on the WHO’s numerical cut-off of five or more concomitant medications [13], with prescription coverage calculated through DDDs and evaluated within a single therapeutic class (ATC fourth level). The application of three different operational definitions resulted in a significant variability of prevalence estimates (range: 20.47–39.98%, depending on the time window considered). Prevalence estimates were inversely proportional to the coverage period required for polypharmacy evaluation. Likewise, the choice of a higher medication exposure to calculate prevalence estimates, such as ≥10 medications [16], was associated with lower prevalence estimates across definitions (range 1.52–10.55%), as expected. Epidemiologists should carefully consider the choice of an operational definition of polypharmacy when calculating prevalence estimates in large population studies. By highlighting a dose–response association of all the definitions of polypharmacy with hospitalization risk, our analysis did not seem to suggest that one definition is better than another. All in all, the goal of obtaining a single and standardized operational definition that can override the influence of various factors, such as numerical cut-offs or time windows, to provide homogeneous estimates in any setting appears currently unfeasible.

Polypharmacy is frequently used as a proxy to characterize patients based on their clinical severity [17] or need for a medication review process [18]. In this regard, the definition DDD ≥ 60 was able to select a cohort of subjects characterized by a higher clinical complexity compared to the others, mainly due to the exclusion of drugs not used for chronic treatments. In addition, while all definitions were confirmed to be associated with a statistically significant increased risk of all-cause hospitalization, DDD ≥ 60 was associated with a lower variability in the estimates over different time windows and showed the absolute lowest risk estimates compared with the other definitions. We can assume that the DDD ≥ 60 definition implicitly encompasses a qualitative dimension by considering polypharmacy in the context of concurrent chronic treatments, necessary to manage conditions that, if not properly controlled, could increase the risk of adverse clinical outcomes. However, this remains a quantitative-only approach. Ideally, the numerical cut-off should be accompanied by the distinction between appropriate and inappropriate treatments. In fact, even if in patients with multiple comorbidities, an elevated number of drugs may be justifiable, and potentially inappropriate drugs still represent a major polypharmacy-related burden [19]. Clearly, such an assessment would require a case-by-case approach and is not applicable for epidemiological estimates on large population data.

Although polypharmacy can be problematic rather than often inappropriate, the assessment of prescribing appropriateness needs to extend beyond the number of drugs prescribed and consider other co-existing medical conditions [20]. This makes polypharmacy management a challenge for health systems worldwide [21].

Patients are frequently under the care of various healthcare professionals, including general practitioners, specialists, nurses, and community pharmacists. This often results in a clinical approach that overly emphasizes the treatment of individual conditions, neglecting a comprehensive assessment of the patient’s overall health status.

As a result, medication adherence can be often compromised, with patients struggling to follow multiple regimens, which leads to worsened health outcomes and reduced quality of life. Given that medication-taking is a complex behavior involving several steps by patients, an upscaling of communicative efforts to facilitate shared decision-making and treatment adherence in many patients exposed to polypharmacy is essential.

A more effective identification of subjects at higher risk of adverse events and deserving a recurrent medication review would be the best solution to minimize negative outcomes associated with polypharmacy. General practitioners (GPs) are increasingly aware of the importance of deprescribing. Despite being well-positioned to implement this process due to frequent contacts and unique patient–physician relationships, challenges remain, including limited time and resources and the need for targeted education and clear guidelines to improve confidence and knowledge. The choice of clinically effective polypharmacy definitions could facilitate the recognition of inappropriate polypharmacy conditions, which if resolved or minimized could also reduce healthcare expenditures.

### 4.2. Strengths and Limitations

Our cohort included over 430,000 adults, ensuring robust and consistent results, also allowing for the evaluation of polypharmacy in the adult non-older population. However, some limitations must be considered due to the administrative nature of available data. First, the intake of unreimbursed drugs, as well as over the counter (OTC) medications, supplements, and herbal products, could not be assessed. While the inclusion of non-prescription products in the evaluation of polypharmacy is still debated [19], a widespread consumption is reported in Italy [22]. Therefore, the actual prevalence of polypharmacy might be underestimated. Second, the use of DDDs for prescription coverage calculations may not accurately reflect the actual prescribed doses; thus, therapy durations might have been misestimated in some cases. Third, another limitation of this study is the reliance on administrative health databases, which do not provide detailed clinical patient information. This limitation may impact the ability to assess the appropriateness of prescribed therapies and understand the complete clinical context of each patient’s medication regimen, leaving potential associations between clinical parameters and polypharmacy unexplored.

## 5. Conclusions

Our study results add to the growing evidence that polypharmacy should be evaluated in two different but coexisting contexts. The clinical evaluation of polypharmacy cannot be based on a numerical-only operational threshold, even if an enhanced knowledge of differences between operational definitions could guide stakeholders to better quantify risks associated with this condition in real-world clinical practice. On the other hand, in the epidemiological evaluation of polypharmacy, the goal of obtaining a single, standard, universal, and validated methodology of polypharmacy assessment appears to be currently unfeasible. Nevertheless, different operational definitions can successfully be used in the analysis of administrative data, resulting in different prevalence estimates but consistently highlighting a dose-dependent increase in all-cause hospitalization risk. Future strategies could leverage artificial intelligence and machine learning applications in healthcare administrative databases. These technologies could efficiently identify polypharmacy patients, making them eligible for medication review and deprescribing activities when necessary. In conclusion, a refinement of the approaches for polypharmacy evaluation is needed to address the challenging identification of exposed subjects more vulnerable to potential harm and to avoid related negative outcomes.

## Figures and Tables

**Figure 1 pharmacy-13-00015-f001:**
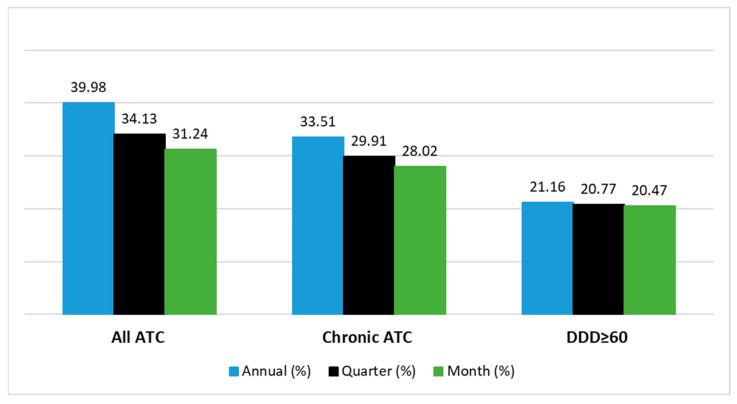
Prevalence of patients in polypharmacy (5 or more drugs). Monthly and quarterly measures of polypharmacy over the year are reported as the maximum number of dispensed drugs between months and quarters.

**Figure 2 pharmacy-13-00015-f002:**
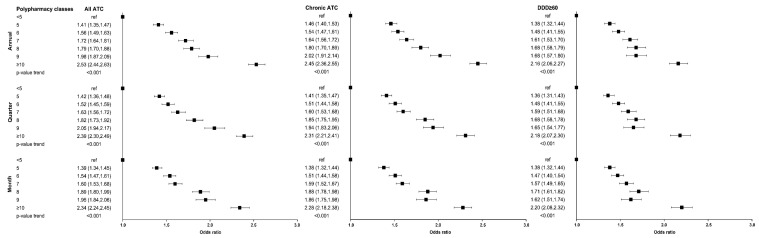
Adjusted estimates (odds ratios (ORs) and 95% confidence intervals [LCL: lower confidence limit; UCL: upper confidence limit]) for the association between polypharmacy classes and the risk of hospitalization for all causes.

**Table 1 pharmacy-13-00015-t001:** Operational polypharmacy definitions applicable to administrative databases.

Definition	Reference	Criteria
All ATC	WHO (2019) [13]	All ATC 4th-level prescriptions were included
Chronic ATC	Kardas et al. (2021) [4]	All ATC 4th-level prescriptions were evaluated, and then ATC 2nd- (A01, A06, J01, J02, J05, J06, J07, P03) and 1st-level (D, V) prescriptions were excluded
DDD ≥ 60	Valent (2019) [14]	Only drugs (ATC 4th level) with a cumulative annual DDDs ≥ 60

**Table 2 pharmacy-13-00015-t002:** Prevalence of patients by polypharmacy classes.

	Polypharmacy Classes	Annual		Quarter		Month	
		N	%	N	%	N	%
All ATC	<5	259,047	60.02	284,315	65.87	296,803	68.76
	5	37,530	8.7	37,335	8.65	37,069	8.59
	6	30,396	7.04	28,994	6.72	28,144	6.52
	7	24,443	5.66	22,435	5.20	21,168	4.90
	8	19,424	4.50	16,817	3.90	15,793	3.66
	9	15,235	3.53	12,777	2.96	11,228	2.60
	≥10	45,545	10.55	28,947	6.71	21,415	4.96
Chronic ATC	<5	287,005	66.49	302,541	70.09	310,664	71.98
	5	34,241	7.93	34,177	7.92	33,998	7.88
	6	27,015	6.26	26,213	6.07	25,824	5.98
	7	21,467	4.97	20,057	4.65	19,457	4.51
	8	16,568	3.84	15,056	3.49	14,176	3.28
	9	12,721	2.95	11,050	2.56	9853	2.28
	≥10	32,603	7.55	22,526	5.22	17,648	4.09
DDD ≥ 60	<5	340,283	78.84	341,987	79.23	343,263	79.53
	5	26,752	6.20	26,707	6.19	26,822	6.21
	6	20,116	4.66	20,097	4.66	20,101	4.66
	7	14,691	3.40	14,559	3.37	14,508	3.36
	8	10,289	2.38	10,116	2.34	9963	2.31
	9	7066	1.64	6818	1.58	6648	1.54
	≥10	12,423	2.88	11,336	2.63	10,315	2.39

**Table 3 pharmacy-13-00015-t003:** Characteristics of subjects in polytherapy (5 or more drugs).

	All ATC	Chronic ATC	DDD ≥ 60
Annual	172,573	144,615	91,337
Sex (F, %)	57.85	57.5	54.52
Age (mean, SD)	68.8 (12.5)	70.3 (12.0)	72.6 (10.9)
Number of ATC fourth-level drugs (mean, SD)	8.2 (3.3)	8.7 (3.3)	9.7 (3.5)
Previous hospitalization (%)	45.14	47.53	51.29
Outcome (Yes, %)	11.53	12.31	13.79
Quarter	147,305	129,079	89,633
Sex (F, %)	56.92	56.71	54.34
Age (mean, SD)	70.1 (12.0)	71.1 (11.6)	72.7 (10.8)
Number of ATC fourth-level drugs (mean, SD)	8.6 (3.3)	9.0 (3.4)	9.8 (3.5)
Previous hospitalization (%)	46.99	48.62	51.40
Outcome (Yes, %)	12.12	12.64	13.8
Month	134,817	120,956	88,357
Sex (F, %)	56.39	56.25	54.22
Age (mean, SD)	70.7 (11.8)	71.5 (11.5)	72.8 (10.8)
Number of ATC fourth-level drugs (mean, SD)	8.9 (3.3)	9.1 (3.4)	9.8 (3.5)
Previous hospitalization (%)	47.87	49.17	51.46
Outcome (Yes, %)	12.43	12.83	13.83

**Table 4 pharmacy-13-00015-t004:** Unadjusted and adjusted estimates (odds ratios (ORs) and 95% confidence intervals [LCL: lower confidence limit; UCL: upper confidence limit]) for the association between polypharmacy (5 or more drugs versus less than 5 drugs) and the risk of hospitalization for all causes.

		All ATC			Chronic ATC			DDD ≥ 60		
		OR	LCL	UCL	OR	LCL	UCL	OR	LCL	UCL
Annual (%)	Crude	2.75	2.69	2.82	2.77	2.71	2.84	2.70	2.64	2.76
	Adjusted	1.81	1.77	1.86	1.77	1.72	1.82	1.60	1.56	1.64
Quarter (%)	Crude	2.71	2.65	2.77	2.71	2.65	2.78	2.68	2.62	2.75
	Adjusted	1.72	1.68	1.77	1.68	1.64	1.73	1.58	1.54	1.63
Month (%)	Crude	2.69	2.63	2.75	2.69	2.63	2.76	2.68	2.62	2.74
	Adjusted	1.68	1.64	1.73	1.65	1.60	1.69	1.58	1.53	1.62

## Data Availability

The raw data supporting the conclusions of this article will be made available by the authors on request.

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
