# Peer review of "Operational Definitions of Polypharmacy and Their Association with All-Cause Hospitalization Risk: A Conceptual Framework Using Administrative Databases"

_pharmacy, 2025, doi:10.3390/pharmacy13010015_

Round 1

Reviewer 1 Report

Comments and Suggestions for Authors

I read with interest the paper titled "Operational Definitions of Polypharmacy and Their Association with All-Cause Hospitalization Risk: A Conceptual Framework Using Administrative Databases"

1. The aim described in the end of the introduction as "recapitulating and describing the variability of different operational polypharmacy definitions" is more an objective for a systematic review. This review was already performed in the past for other authors that you cite as reference 9. If the objective is that one, an update to this systematic review is more appropriate. 

2. Then, part two of the objective is to use real world data for assess hospitalization using real world data. I wonder why the RWD is from 2018. More recent data is not available? Whats the rationale to choose that period?

3. Why the age of patients were limited to older than 40?

4. Table 1. ATC 3rd level is composed by one letter two numbers and 1 letter, so the information in the table should be revised. e.g. A01A, A01B)

Author Response

1. Summary

Thank you very much for taking the time to review this manuscript. Below, you will find our detailed responses to the comments, with the corresponding revisions highlighted in red in the re-submitted manuscript file (please see the attachment).

2. Point-by-point response to Comments and Suggestions for Authors

Comment 1: The aim described in the end of the introduction as "recapitulating and describing the variability of different operational polypharmacy definitions" is more an objective for a systematic review. This review was already performed in the past for other authors that you cite as reference 9. If the objective is that one, an update to this systematic review is more appropriate.

Response 1: We thank the reviewer for pointing this out.

To better clarify our aim, we have revised the manuscript as follow:

“In this context, building on published literature, we aimed to identify a set of operational definitions of polypharmacy applicable to administrative databases and to assess their association with the risk of all-cause hospitalization using real-world data.”

This revision can be found in the revised manuscript on page 2, lines 63–66.

Additionally, we have updated the aim stated in the abstract to ensure consistency:

“Building on published literature, this study aimed to identify a set of operational definitions of polypharmacy applicable to administrative databases and to assess their association with all-cause hospitalization.”

This revision is located on page 1, lines 13–15.

Comment 2: Then, part two of the objective is to use real world data for assess hospitalization using real world data. I wonder why the RWD is from 2018. More recent data is not available? Whats the rationale to choose that period?

Response 2: We thank the reviewer for this important question.

The reason for using 2018 data is that the EDU.RE.DRUG study dataset, which we had access to, covered the period from 2014 to 2018.

Since our analysis does not focus on specific drug utilization patterns or temporal trends in polypharmacy prevalence, we believe that the choice of a specific timeframe has a limited impact on the validity of our findings. When assessing the epidemiological differences that emerge from applying different operational definitions, we felt that the selection of a recent period of analysis was not a priority, given the purely methodological nature of our work.

Comment 3: Why the age of patients were limited to older than 40?

Response 3: We thank the reviewer for raising this point.

The selection of patients aged 40 or older is due to the prevalence of polypharmacy, which is particularly common in adults aged 65 and older. Extending the inclusion of adults aged 40–64 years was an informed decision in the EDU.RE.DRUG study, enabling the authors to broaden the investigation to a wider range of patients.

Comment 4: Table 1. ATC 3rd level is composed by one letter two numbers and 1 letter, so the information in the table should be revised. e.g. A01A, A01B)

Response 4: We thank the reviewer for pointing out this error.

We have revised the table to correctly indicate the ATC codes with one letter and two numbers as second-level ATC codes (ATC 2nd level).

This change can be found in the manuscript on page 3, Table 1.

Reviewer 2 Report

Comments and Suggestions for Authors

Dear Authors, I really appreciated you r paper. To improve it, please find below my comments:

1. I suggest to clarify in the introduction that the problem is a consequence of the specialization.

2. You might add some raws about the compliance 

3. Do you have some data about the drugs that are most used?

4. It would be helpful for the reader to clarify who has the responsibility and the competences to reduce the number of drugs? The general practitioner of the most relevant sickness.

5. The paragraph 2.2 is very short. you can include it at the end of paragraph 2.1 or at the beginning of paragraph 2.3

6. I suppose you consider the active principle of both label and equivalent drugs.

7. Why do you consider only hospitalization over 7 days? Please clarify

8. Do you think that real world evidence and artificial intelligence could help to reduce the problem? You can say something in the conclusions.

Author Response

1. Summary

Thank you very much for taking the time to review this manuscript. Below, you will find our detailed responses to the comments, with the corresponding revisions highlighted in red in the re-submitted file (please see the attachment).

2. Point-by-point response to Comments and Suggestions for Authors

Comment 1: I suggest to clarify in the introduction that the problem is a consequence of the specialization.

Response 1: We thank the reviewer for this comment.

In response, we have added a brief sentence in the introduction, which can be found on page 1, lines 37-39:

"This is exacerbated by an increasing medical specialization within healthcare systems, leading to multiple treatment regimens, often without any reconciliation process.”

Comment 2: You might add some raws about the compliance

Response 2: We thank the reviewer for the valuable suggestion.

We have now added some lines in the discussion regarding medication adherence, which can be found on page 9-10, lines 284-288 of the revised manuscript:

"As a result, medication adherence can often be compromised, with patients struggling to follow multiple regimens, which leads to worsened health outcomes and reduced quality of life. Given that medication-taking is a complex behavior involving several steps by patients, an upscaling of communicative efforts to facilitate shared decision-making and treatment adherence in many patients exposed to polypharmacy is essential."

We hope this response helps clarify the point.

Comment 3: Do you have some data about the drugs that are most used?

Response 3: Thank you for your question.

We have added the data on the top five most frequently prescribed active ingredients in the analyzed cohort to the manuscript. These details can now be found in the Results section, page 4, lines 160-164:

'The top five most frequently prescribed drugs/drug classes in this cohort were proton pump inhibitors (149,574 prescriptions), vitamin D and analogues (110,845 prescriptions), 3-hydroxy-3-methylglutaryl-CoA (HMG-CoA) reductase inhibitors (94,384 prescriptions), combinations of penicillins, including beta-lactamase inhibitors (92,486 prescriptions), and fluoroquinolones (80,205 prescriptions). (see Supplementary Table XX ). '

Comment 4: It would be helpful for the reader to clarify who has the responsibility and the competences to reduce the number of drugs? The general practitioner of the most relevant sickness.

Response 4: We would like to thank the reviewer for this additional insight.

We fully share the view that general practitioners are most suited to reduce the number of inappropriate or unnecessary medications. In response, we have added the following lines to the conclusions, on pages 9 and 10, lines 291-295:

"General practitioners (GPs) are increasingly aware of the importance of deprescribing. Despite being well-positioned to implement this process due to frequent contacts and unique patient-physician relationship, challenges remain, including limited time and resources, and the need for targeted education and clear guidelines to improve confidence and knowledge."

We hope this addresses the reviewer’s suggestion.

Comment 5: The paragraph 2.2 is very short. you can include it at the end of paragraph 2.1 or at the beginning of paragraph 2.3

Response 5: We thank the reviewer for the suggestion.

In response, we have merged paragraph 2.2 with paragraph 2.1, which has now been retitled "2.1. Data sources and study population." This can be found on page 2, line 68.

Comment 6: I suppose you consider the active principle of both label and equivalent drugs.

Response 6: We thank the reviewer for the question.

The EDU.RE.DRUG study dataset included information on the dispensing of both label and equivalent drugs. Therefore, we confirm that our analyses were conducted based on all active ingredients, regardless of whether the drug was a branded specialty or an equivalent drug.

Comment 7: Why do you consider only hospitalization over 7 days? Please clarify

Response 7: Thank you for your question.

We apologize for the lack of clarity in the original manuscript, as the wording was not sufficiently explicit. In the main analysis, all hospitalizations were considered, while hospitalizations lasting more than 7 days were only considered in the sensitivity analysis, as reported in Supplementary Table 4.

We have amended the manuscript to clarify this by specifying 'any hospitalization' on line 103, page 3, and adding 'only hospitalizations longer than 7 days' on lines 226-227, page 8.

Comment 8: Do you think that real world evidence and artificial intelligence could help to reduce the problem? You can say something in the conclusions.

Response 8: We thank the reviewer for suggesting this interesting perspective.

We have added two sentences in the conclusion regarding the potential use of AI in this context, which can be found on page 10, lines 324-328:

'Future strategies could leverage artificial intelligence and machine learning applications in healthcare administrative databases. These technologies could efficiently identify polypharmacy patients, making them eligible for medication review and deprescribing activities when necessary.'
